# Cross-Sectional and Longitudinal Effects of PCB Exposure on Human Stress Hormones in the German HELPcB Surveillance Program

**DOI:** 10.3390/ijerph17134708

**Published:** 2020-06-30

**Authors:** Petra Maria Gaum, Viktoria Stefanie Vida, Thomas Schettgen, André Esser, Thomas Kraus, Monika Gube, Jessica Lang

**Affiliations:** 1Institute for Occupational, Social and Environmental Medicine, RWTH Aachen University, Pauwelsstraße 30, 52074 Aachen, North Rhine Westphalia, Germany; viktoria.vida@rwth-aachen.de (V.S.V.); tschettgen@ukaachen.de (T.S.); anesser@ukaachen.de (A.E.); tkraus@ukaachen.de (T.K.); monika.gube@staedteregion-aachen.de (M.G.); jlang@ukaachen.de (J.L.); 2Health Authority of the City and Area of Aachen, Trierer Straße 1, 52070 Aachen, Germany

**Keywords:** polychlorinated biphenyls, stress hormones, cortisol, DHEAS, adults, HELPcB

## Abstract

Multiple pathological associations are attributed to PCBs (polychlorinated biphenyls). Research shows a positive association of PCBs with dehydroepiandrosterone sulfate (DHEAS) concentration but the results on the stress hormone cortisol have been inconsistent so far. This study is the first to examine not only the cross-sectional but also the longitudinal effects of PCB exposure on the stress hormones DHEAS and cortisol. Over a period of three years, 112 former employees occupationally exposed to PCBs were tested for their body burden with different types of PCBs (lower and higher chlorinated, dioxin-like and hydroxylated) and for their stress hormone concentration. Highly exposed employees showed a significantly higher risk for higher DHEAS values. Multiple linear regression analysis revealed a positive relationship between the exposure to lower chlorinated PCBs and DHEAS. Mixed models also revealed a significantly positive correlation between lower chlorinated PCBs with DHEAS when controlled for a cross-section. However, an effect for cortisol was not found. These results suggest a causal pathophysiological relationship between PCB exposure and DHEAS concentration, but not with cortisol. The health consequences of high DHEAS concentrations are discussed.

## 1. Introduction

Polychlorinated biphenyls (PCBs) are a group of synthetically produced hazardous chemicals, whose negative consequences on human health and the environment have become visible decades after their intensive production and use [1]. The present study analyzes potential adverse health effects of PCBs on the individuals’ stress response system and focuses on the influence of PCBs on human stress hormones. Prior studies mostly report positive associations between PCBs and dehydroepiandrosterone sulfate (DHEAS) in humans, but no or inconsistent findings with cortisol [2]. This longitudinal study addresses both, DHEAS and cortisol to find causal relations.

Due to their adverse health impact, in 1995, the manufacture, import, export and sale of PCBs and products and equipment containing PCBs have been banned worldwide by the UNEP (United Nations Environment Programme) [3]. In Germany, the use of PCBs was already banned in 1989. Prior to the ban, these toxic chlorine compounds were widely used in electrical capacitors and transformers, as well as in hydraulic fluids, lubricants, pesticides and plasticizers [4]. Of the 209 theoretically possible PCB congeners, 130 have been detected in the environment. The PCB congeners, which all have the same basic structure, differ from each other in the number and position of chlorine substituents [5]. Based on their degree of chlorination (lower vs. higher chlorination) and their structural properties (non-dioxin-like vs. dioxin-like), PCB congeners can be classified into three categories: lower chlorinated PCBs (LPCBs; ≤ 5 chlorine atoms), higher chlorinated PCBs (HPCBs; >five chlorine atoms), and dioxin-like PCBs (dlPCBs) that include lower- as well as higher-chlorinated congeners but with no or only one chlorine atom in the ortho positions. Furthermore, this study also addresses a fourth group, the hydroxylated PCB metabolites (OH-PCBs). Due to the high persistence and bioaccumulative properties of the parent-PCB-congeners, PCBs remain in the environment and are still detectable in the blood of the general population [6]. According to Carpenter, the majority of the population in developed countries is exposed to PCBs through food intake (i.e., HPCBs) [7]. Furthermore, the place of living affects the exposure with PCBs, particularly in homes near hazardous waste sites [7]. Work-related contact with PCB-contaminated material represents an additional exposure in the form of inhalation as well as cutaneous PCB intake (i.e., LPCBs and dlPCBs with lower degree of chlorination) [8]. Past research has also described differential health effects for the different types of PCBs and OH-PCBs [7,9].

PCBs are attributed to reproductive toxicity [10], immunotoxicity [11] and neurotoxic effects [12]. Moreover, hormonal changes in humans after PCB exposure were found in prior studies, for instance with thyroid hormones [13] or the sex hormone free testosterone [14]. In addition, a few studies focused on stress hormones in humans after PCB exposure, but these studies report inconsistent and sometimes contradictory results. For instance, Persky et al. found an inverse association between increased PCB body burden and the DHEAS concentration after the occupational PCB exposure of female workers during menopause [15]. However, a subsequent study found a positive association between PCB exposure and the DHEAS concentration in male employees [2]. Sun et al. also report a significant increase in DHEA concentration with increased PCB body burden in men [16]. High DHEAS inhibits genetically programmed cell death [17], and this enhances the risk for cancer. PCBs were classified as human carcinogens by the IARC (International Agency for Research on Cancer) in 2013 [18]. Related to cortisol, no significant results for the effect of PCBs have been found in humans so far [2,15]. Possible reasons of these inconsistent findings may be the cross-sectional design of prior studies. PCBs may have different effects on human hormones depending on how long ago the exposure occurred. For example, in an earlier longitudinal study we found an interaction with time for the association between PCBs and free thyroxine [13]. Cross-sectional study designs cannot detect changes in associations over time and report only a snapshot, which can result in inconsistent findings in various articles.

The aim of this study is therefore to enable a causal interpretation of the relationship between PCB body burden and the stress hormone concentration by conducting a longitudinal study with a repeated assessment of both PCBs and stress hormones. According to the above-mentioned studies, a positive association between PCB body burden and the DHEAS concentration is postulated. Furthermore, the association between the PCB body burden and the serum cortisol concentration is tested. This study consists of three parts to test the postulated hypothesis and the research question. In the first part, we compare categorized and mean stress hormone concentrations of higher exposed participants with those participants showing a PCB concentration at the level of background burden. We hypothesize that the higher exposed participants show a higher probability for DHEAS concentrations above the reference range (Hypothesis 1a). Research question 1a considers differences between higher and background-burdened participants according to abnormal cortisol concentrations above or under the reference range. We further expect a higher mean DHEAS concentration for the higher PCB-burdened participants (hypothesis 1b) and we investigate mean differences in cortisol (research question 1b). In the second part, for a better comparability with prior cross-sectional studies, the linear cross-sectional effects of PCB exposure on stress hormone concentrations are analyzed. Again, we expect a positive association between PCB body burden and DHEAS concentration (hypothesis 2). Furthermore, the association between PCBs and cortisol will be tested (research question 2). The third part analyzes the linear relation between PCB body burden and the stress hormone concentrations longitudinally while controlling for the different sampling time points. Hypothesis 3 postulates a positive association between PCBs and DHEAS continuously over all three sampling time points. Research question 3 focusses on the longitudinal associations between PCBs and cortisol.

## 2. Materials and Methods

### 2.1. Study Design

The participants were observed with a three-wave longitudinal within-subjects-design. There was a one-year time lag between each sampling time point (t1–t3). The data were collected as part of the HELPcB surveillance program (Health Effects in high Level exposure to PCBs). For further details, see Kraus et al. [4]. The HELPcB program was approved by the local Ethics Commission of the Medical Faculty of the RWTH Aachen University, Germany (no. EK 176/11).

### 2.2. Study Population

The HELPcB cohort consisted of 300 former employees of a recycling company for recycling industrial capacitors and transformers and employees’ relatives. These workers were PCB exposed during their work by cleaning old capacitors and transformers to prepare them for recycling. Of these 300 participants, 188 were excluded because of the following criteria: 131 persons (43.7%) did not participate at all three sampling time points. Since past research has provided evidence for gender-specific differences in DHEAS concentration [19] and cortisol production and metabolism [20], we excluded females who were underrepresented in our sample (*N* = 49, 16.3%). Eight further participants (2.67%) were excluded, because they were taking cortisol-related drugs at the time points of investigation. Thus, the final study sample consists of 112 men occupationally exposed to PCBs with a mean age of 47.3 years (SD = 12.5).

### 2.3. Data Collection

#### 2.3.1. Polychlorinated Biphenyls and Hydroxylated Biphenyls

The exposure of PCBs and their hydroxylated metabolites (OH-PCBs) was measured in plasma via human biomonitoring. A detailed description of the PCB and OH_PCB analyses including method validation is in Appendix A.

In the analysis, the indicator congeners of LPCBs (PCB28, PCB52, PCB101) as well as HPCBs (PCB138, PCB153, PCB180) were determined and sum values of these six congeners were generated. Twelve dlPCBs were also measured, but only the eight mono-ortho dlPCBs (PCB105, PCB114, PCB118, PCB123, PCB156, PCB157, PCB167 and PCB189) could be detected in more than 20% of the participants and included in the analysis. For generating the fourth category, the concentration of 13 congeners of OH-PCBs were summed up (3-OH-CB28, 4-OH-CB61, 4-OH-CB76, 4-OHCB101, 4-OH-CB107, 4-OH-CB108, 3-OH-CB118, 3-OH-CB138, 4-OH-CB146, 3-OH-CB153, 4-OH-CB172, 3-OH-CB180 and 4-OH-CB187). All considered dlPCBs have the same toxic equivalency factor; thus, it was not controlled for it [21]. A description of the PCB sum variables is in Table 1 and for each included congener in Appendix B.

Since PCBs accumulate in fatty tissue due to its lipophilic property, the PCB values were lipid adjusted. The formula of Bernert et al. was used to determine the total blood lipids for each participant (total lipids = (2.27 * total cholesterol) + triglycerides + 62.3 mg/dL)), and the PCB level was divided by these total blood lipid values [22]. Due to their hydroxylation and the subsequent conjugation with glucuronide or sulfate in the liver, OH-PCBs are not as lipophilic as the parent compounds and thus they were not lipid adjusted.

#### 2.3.2. Stress Hormones

The concentrations of DHEAS and cortisol were measured in the blood plasma. In the internal laboratory procedure, the DHEAS and cortisol samples were incubated with a hormone-specific antibody. An immune complex was formed, and, in the next step, this was transferred to the solid phase. In the measuring cell, the micro particles are fixated by magnetic action. A chemical luminescence emission is induced by applying a voltage and the DHEAS and cortisol content is measured with a photomultiplier. The individual laboratory-specific reference range for DHEAS varied between 1.40 and 8.00 μmol/L and for cortisol between 171 and 536 nmol/L. A detailed description of the final study population as well as reference values for the age-related PCB levels of the general population and hormone concentrations are presented in Table 1.

#### 2.3.3. Statistical Analyses

Prior to hypothesis testing, the additional influence of confounding variables was investigated. Therefore, a directed acyclic graph was generated with the online tool DAGitty version 3.0 [24]. The graph is illustrated in Figure A1 (Appendix C) in the supplement. Relevant confounding factors were age, liver function and smoking behavior of participants that were statistically controlled in further analyses. Age is associated with increased PCB body burden [25] and with a reduction in the hormone secretion of DHEAS [26]. Equally decisive for the individual extent of internal PCB exposure is liver function, as it is also responsible for the metabolism of PCBs. Liver function is also important for the metabolism of stress hormones, because stress hormones are metabolized in the liver. We operationalized liver function by the plasma protein albumin [27]. Finally, we controlled for the smoking behavior of study participants as the third confounding variable, because smoking is involved in the metabolism of PCBs [28]. At the same time, nicotine is a strong activator of the hypothalamic pituitary adrenal axis (HPA) and is permanently associated with attenuated cortisol reactions [29]. As a next step, we analyzed the correlations between all types of PCBs with the outcome variables using Spearmans’ rank correlation coefficients.

Then, in the first part, the focus was on the comparison of higher-burdened participants with participants who have a PCB burden at background level. Therefore, dichotomous variables were generated for each PCB sum variable. The decisive factor was the 95th percentile of a reference population without additional work-related PCB exposure [6,23]. Participants with a PCB body burden above the 95th percentile were categorized as higher burdened and all others as background burdened. Detailed information of the two built groups is in Table 2. In the absence of reference values from the general population for OH-PCBs, no dichotomous variables can be created and no risk analyses and analyses of mean differences can be carried out for OH-PCBs. To test hypothesis and research question 1a, two dichotomous variables were generated for abnormal stress hormone concentrations. In the first variable for abnormal low hormone concentrations, participants with stress hormone concentrations under the reference range were coded with “1” and participants with hormone concentrations in the reference range with “0”. In the second variable, abnormal high hormone concentrations were considered and participants with stress hormone concentrations above the reference range were again coded with “1” and participants with concentrations in the reference range again with “0”. With the dichotomous PCBs and hormone variables, odds ratios for risk analysis were calculated. For analyzing hypothesis 1b and answering research question 1b and the mean differences in stress-hormone concentrations, an analysis of covariance (ANCOVA) was performed with dichotomous PCB variables as predictor and the continuous stress hormone variables as outcome variables. In the second part, the focus was on the cross-sectional linear association between PCB body burden and the stress hormone concentrations. Therefore, a multiple linear regression analysis was performed for each sampling time point with the PCB sum variables as predictor and DHEAS as well as cortisol as criterion variables. For a first insight into longitudinal associations, correlations between PCBs and the outcome DHEAS and cortisol over different sampling time points were analyzed. The third part, focusing on the longitudinal effect of PCBs on stress hormones, was calculated by using mixed effect models with controlling for sampling time points as described by Baayen et al. [30].

All analyses in part one and two were performed with SPSS 25 (IBM, Armonk, NY, USA) for Windows [31], and, for the analyses in part three, the statistical software R version 3.5.0 [32] and RStudio version 1.1.383 (RStudio Inc., Boston, MA, USA) [33] with the package lme4 [34] were used. All hypotheses were tested one sided, since directed hypotheses were postulated and all research questions two-sided, because no direction of the effect could be expected based on the prior research. For both hypotheses testing and answering the research questions, a significance level of *p* = 0.05 was used. Since all PCB variables are not normally distributed, they were transformed using the natural logarithm to approximate them to the normal distribution.

## 3. Results

The correlations between all types of PCBs and the relevant outcome variables at each sampling time point are presented in Table 3. There are positive correlations between LPCB and DHEAS at all sampling time points and between dlPCBs and OH-PCBs with DHEAS at t2. Between PCBs and cortisol there were no significant correlations. The correlations between the several PCB congeners with the relevant outcome variables are presented in the supplementary table, Table A1.

In the first part of this study, participants with a higher PCB burden were compared with background-burdened participants. It was tested whether participants with higher a PCB burden had a higher probability for elevated DHEAS concentrations than background burdened (hypothesis 1a) and whether there are differences in the probability for elevated or reduced cortisol concentrations (research question 1a). Participants with higher exposure in LPCBs and dlPCBs had a higher risk of elevated DHEAS concentrations compared to the background-burdened participants, but only at t2 (Table 4). With regard to cortisol, there was no higher risk of abnormal values, neither elevated nor reduced concentrations, in the higher PCB-exposed group compared to the background-burdened group (Table 4). The comparison of the mean concentrations of DHEAS (hypothesis 1b) and cortisol (research question 1b) revealed that participants with higher LPCB exposure had a significantly higher mean DHEAS concentration than participants with a normal LPCB body burden at t2 and t3 (Table 5). Furthermore, participants with a higher body burden of dlPCBs showed significant higher DHEAS concentrations at t2. Regarding the cortisol concentration, no significant mean differences between the higher and background-burdened participants can be found in all three sampling time points. According to these results, hypotheses 1a and 1b can be partially confirmed, but research questions 1a and 1b could not be answered.

The second part focused on the linear association between the PCB body burden and the concentrations of DHEAS (hypothesis 2) and cortisol (research question 2) for each sampling time point. These cross-sectional analyses showed one significant positive correlation between LPCB body burden and the DHEAS concentration for t2 (Table 6). No significant correlations were found for HPCBs, dlPCBs and OH-PCBs and for all other sampling time points. The cortisol concentration was not significantly associated with any type of PCB body burden. The results partially support the postulated hypothesis and the research question could not be answered.

In the last part, the linear association between PCB body burden with DHEAS (hypothesis 3) and cortisol concentration (research question 3) were tested over time controlled for the influence of the sampling time point. The results of the mixed effect models confirmed the correlation between PCB body burden and the DHEAS concentration for LPCBs (Table 7). For HPCBs, dlPCBs and OH-PCBs no significant correlations were found with DHEAS. As in the analyses before, no association could be found between any type of PCB and cortisol. The postulated hypotheses, again, were partially confirmed for DHEAS, but the research question according to cortisol could also not be answered under control for the sampling time points.

## 4. Discussion

The aim of this study was to investigate the effects of PCBs on the stress hormones DHEAS and cortisol. According to the literature, a positive association between PCB body burden and DHEAS concentration was postulated. Furthermore, undirected research questions were formulated for the associations between PCBs and cortisol because of prior inconsistent findings. To test the postulated hypotheses and research questions, this study was structured in three parts.

Higher-burdened participants in LPCBs and dlPCBs have an approximately two- and three-fold higher risk for elevated DHEAS concentrations compared to background-burdened participants. The mean DHEAS concentration was also higher in the higher exposed group, but only for LPCBs and dlPCBs. However, the findings concerning the mean differences must be interpreted with caution, because there was a variance inhomogeneity, which could have had an impact on the effect size. According to cortisol, no differences in risk or mean concentrations were found between the higher- and background-burdened group. The linear association between PCB exposure and stress hormone concentration was examined for each sampling time point. As in the first part, PCB exposure only affected the DHEAS concentration, but not the cortisol concentration. At t2 an increase of the DHEAS concentration could be observed with an increase of LPCB body burden. The same positive association between LPCB exposure and the DHEAS concentration was found in the third part when controlling for the sampling time points. As before, there was also no effect related to cortisol in the longitudinal analyses. The results of this study confirmed previous studies that also found a positive association between PCB exposure and DHEAS concentration, but none between PCB exposure and cortisol concentration [2,15,16].

We found the clearest effects for LPCBs and DHEAS. Interestingly, LPCBs have a shorter half-life than HPCBs and dlPCBs, which is the reason why they have often been reported to cause less damage to health [35]. However, studies of work-related exposures show increased interest in LPCBs [36]. The relevance of LPCBs for work-related exposure results from the non-food-related intake of PCBs (i.e., dermal or inhalative), which is mainly the path of exposure of LPCBs, and from the exposure material itself. Many PCB mixtures commercially used in Germany, such as Clophen A30, Clophen A40 or Aroclor 1242, contained high proportions of LPCBs [37]. Our study cohort consists of former workers of a recycling company with an occupational PCB exposure. In this study, LPCBs consist of congeners with less than six chlorine atoms. It may be that the degree of chlorination is important for stress-hormone-related health effects. Additional analyses in this study support this. As can be seen in Table A2 (Appendix D) in the supplement, there are positive associations of all considered PCB congeners with less than six chlorine atoms and DHEAS. According to these associations, it can be concluded that the degree of chlorination may be important in the elevation of DHEAS levels after PCB exposure. This may also be an explanation for the inconsistent findings about the associations of PCB exposure and DHEAS in research. Many studies use different types of PCBs for testing the effects of PCBs on stress hormones and thus result in inconsistent findings, as different PCB congeners might have different effects on stress hormones.

As reported before, in this study, PCBs show an effect on plasma DHEAS concentration, but not on cortisol concentration. This result is in line with prior research, but the mechanism is not clearly described. Both steroid hormones are synthesized and released in the adrenal cortex. However, they are produced within the adrenal cortex in two different regions, cortisol in the zona fasciculata and DHEA as well as DHEAS in the zona reticularis [38]. The difference of origin concerning the morphological zones could be a possible explanation that only DHEAS is affected by PCBs and not cortisol. It might be that PCBs only affect the zona reticularis in which DHEAS is produced.

The health consequences of an elevated DHEAS concentration after PCB exposure are difficult to predict due to the not well-known physiological mechanisms and pathological effects of DHEAS abnormalities [17]. However, DHEAS, as an antagonist of cortisol, is mostly attributed with positive effects, such as the improvement of immune function or the stimulation of muscle and bone formation [39]. Maninger et al. described that DHEAS inhibits genetically programmed cell death (apoptosis), which is necessary for subsequent cell proliferation [17]. However, if apoptosis does not occur after a genotoxic lesion, in vitro studies have shown a growth advantage for tumor-promoting and toxicologically influenced cells [40]. A relevant effect of DHEAS might be its apoptosis-inhibiting property. Higher DHEAS concentrations after PCB exposure may be a mechanism for the development of tumors and cancer after PCB exposure [18]. The results of this study give first hints of such an underlying mechanism and recommend further investigation of DHEAS and the general physiological mechanisms in the case of PCB-burdened cells focused on tumor development. Next to somatic consequences in case of higher DHEAS concentrations, mental health problems can also occur. Uh et al. found a non-linear correlation between the DHEAS concentration and the occurrence of depression of varying severity [41]. Furthermore, Lee et al. considered DHEAS primarily as a biomarker for manic symptoms and cognitive performance [42]. Future research should also investigate the role of elevated DHEAS as a pathophysiological mechanism for depression after PCB exposure [43].

A particular strength of this work is its longitudinal design. This allows a causal interpretation of the results and strengthens the described associations. Furthermore, mixed effect models were used to maximize statistical power and to reduce potentially distorting, inter-individual changes by considering random effects of the sampling time points. Because of the male study cohort, the generalizability of the study results are limited in terms of their applicability to PCB-exposed women. However, by using a male study cohort, biases according to gender-differences in stress hormone concentrations were reduced. Gender does influence the amount of PCB burden as well as the concentrations of DHEAS and cortisol [44,45]. The sample of 112 persons represents a well-founded size considering the fact of a longitudinal design and the strict selection criteria used, such as occupational PCB exposure and the exclusion of participants taking cortisol effective drugs. In addition, the use of mixed effect models result in a higher statistical power that reduces biases of the sample size.

## 5. Conclusions

In conclusion, this was the first longitudinal study to investigate the potential exposure effects of PCBs on the stress hormones DHEAS and cortisol in humans. The results reveal a causal positive association between PCB exposure and subsequent DHEAS concentration, but not cortisol concentration. Further research should investigate the mechanism behind the positive association between PCB exposure and DHEAS increase in plasma.

## Figures and Tables

**Table 1 ijerph-17-04708-t001:** Descriptive data of PCBs and stress hormones (*N* = 112).

Variable	Unit	Reference	t1				t2				t3			
			Median	Mean	SD	Min–Max	Median	Mean	SD	Min–Max	Median	Mean	SD	Min–Max
LPCBs	ng/g lipid	0.074 ^a^	0.03	0.43	1.90	<LOD–19.3	0.017	0.3	1.34	<LOD–13.0	0.01	1.93	0.88	<LOD–8.7
HPCBs	ng/g lipid	0.49 ^a,b^	0.37	1.12	1.92	0.04–13.9	0.36	1.06	1.86	0.04–11.8	0.41	1.06	1.66	0.5–11.6
dlPCBs	ng/g lipid	0.001 ^a^	0.08	0.39	0.75	0.01–5.1	0.07	0.37	0.82	0.1–6.3	0.06	0.28	0.49	0.01–3.2
OH-PCBs	µg/L plasma	-	0.81	3.99	7.87	0.1–52.8	0.82	4.11	10.21	0.2–89.7	0.92	2.79	4.37	0.1–26.7
Cortisol	nmol/L	171–536	404.00	416.93	159.50	143.0–892.0	453	443.4	133.94	42.0–769.0	416	432.77	143.08	139.0–846.0
DHEAS	µmol/L	1.4–8.0	6.91	7.76	4.05	0.2–18.5	6.61	7.18	4.03	0.2–22.3	6.11	6.76	3.56	0.1–16.8

Notes: PCBs = polychlorinated biphenyls, LPCBs = lower-chlorinated PCBs, HPCBs = higher-chlorinated PCBs, dlPCBs = dioxin-like PCBs, OH-PCBs = hydroxylated PCBs, DHEAS = dehydroepiandrosterone sulfate, t1–t3 = sampling time point 1–3, SD = standard deviation, LOD = limit of detection. ^a^ See Schettgen et al. [6]; ^b^ See Schettgen et al. [23].

**Table 2 ijerph-17-04708-t002:** Descriptive data of PCB exposure in participants with a PCB background burden and a higher PCB burden.

Variable	Percentile	t1					t2					t3				
		*n*	Median	Mean	SD	Min–Max	*n*	Median	Mean	SD	Min–Max	*n*	Median	Mean	SD	Min–Max
LPCBs	≤95th	28	0.003	0.004	0.002	<LOD–0.01	35	0.004	0.004	0.002	<LOD–0.01	48	0.004	0.004	0.003	<LOD–0.01
	>95th	84	0.08	0.57	2.18	0.01–19.34	77	0.05	0.44	1.60	0.01–13.02	64	0.05	0.34	1.14	0.01–8.72
HPCBs	≤95th	58	0.23	0.23	0.11	0.04–0.47	59	0.22	0.22	0.11	0.04–0.44	51	0.22	0.24	0.13	0.05–0.67
	>95th	54	1.36	2.08	2.43	0.26–13.86	53	1.28	2.01	2.37	0.22–11.79	61	0.98	1.75	2.00	0.12–11.63
dlPCBs	≤95th	48	0.03	0.03	0.02	0.01–0.08	53	0.03	0.03	0.01	0.01–0.07	54	0.03	0.03	0.17	0.01–0.08
	>95th	64	0.29	0.66	0.90	0.02–5.11	60	0.29	0.66	1.05	0.02–6.34	58	0.28	0.51	0.59	0.02–3.15

Notes: PCBs = polychlorinated biphenyls, LPCBs = lower-chlorinated PCBs, HPCBs = higher-chlorinated PCBs, dlPCBs = dioxin-like PCBs, OH-PCBs = hydroxylated PCBs, DHEAS = dehydroepiandrosterone sulfate, t1–t3 = sampling time point 1–3, *n* = frequency, SD = standard deviation, LOD = limit of detection.

**Table 3 ijerph-17-04708-t003:** Cross-sectional Spearmans’ rank correlation coefficients between the PCB variables and DHEAS and cortisol (*N* = 112).

	t1		t2		t3	
	DHEAS	Cortisol	DHEAS	Cortisol	DHEAS	Cortisol
LPCBs	**0.31 ****	−0.06	**0.35 ****	0.08	**0.25 ****	−0.06
HPCBs	0.02	−0.03	0.07	0.03	0.04	−0.11
dlPCBs	0.15	−0.05	**0.19 ***	0.04	0.15	−0.09
OH-PCBs	0.15	−0.08	**0.20 ***	0.06	0.13	−0.15

Note: PCBs = polychlorinated biphenyls; LPCBs = lower-chlorinated PCB; HPCBs = higher-chlorinated PCB; dlPCBs = dioxin-like PCB; DHEAS = dehydroepiandrosterone sulfat; t1–t3 = sampling time point 1–3. ** *p*-value (one-sided) < 0.01; * *p*-value (one-sided) < 0.05. Significant results are in bold.

**Table 4 ijerph-17-04708-t004:** Risk analyses of abnormal DHEAS and cortisol concentration for higher- and background-burdened participants (*N* = 112).

PCB-Exposure		t1			t2			t3		
	PC	%a (*n*)	OR	95%-CI	%a (*n*)	OR	95%-CI	%a (*n*)	OR	95%-CI
DHEAS ↑										
LPCB	≤95th	32.1 (9)			20.0 (7)			29.2 (14)		
	>95th	48.2 (40)	2.0	0.8–4.8	44.7 (34)	**3.2**	**1.3–8.3**	42.2 (27)	1.8	0.8–3.9
HPCB	≤95th	37.9 (22)			29.3 (17)			33.3 (17)		
	>95th	50.9 (27)	1.7	0.8–3.6	45.3 (24)	2.0	0.9–4.4	39.3 (24)	1.3	0.6–2.8
dlPCB	≤95th	35.4 (17)			26.9 (14)			31.5 (17)		
	>95th	50.8 (32)	1.9	0.9–4.1	45.8 (27)	2.3	1.0–5.1	41.4 (24)	1.5	0.7–3.3
DHEAS ↓										
LPCB	≤95th	3.6 (1)			2.9 (1)			2.1 (1)		
	>95th	2.4 (2)	0.7	0.1–7.6	3.9 (3)	1.4	0.1–13.9	4.7 (3)	2.3	0.2–22.9
HPCB	≤95th	3.4 (2)			5.2 (3)			5.9 (3)		
	>95th	1.9 (1)	0.5	0.1–6.1	1.9 (1)	0.4	0.0–3.5	1.6 (1)	0.3	0.0–2.7
dlPCB	≤95th	4.2 (2)			5.8 (3)			5.6 (3)		
	>95th	1.6 (1)	0.4	0.0–4.2	1.7 (1)	0.3	0.0–2.8	1.7 (1)	0.3	0.0–3.0
Cortisol ↑										
LPCB	≤95th	28.6 (8)			22.9 (8)			33.3 (16)		
	>95th	21.7 (18)	0.7	0.3–1.8	21.1 (16)	0.9	0.3–2.4	21.9 (14)	0.6	0.2–1.3
HPCB	≤95th	25.9 (15)			19,0 (11)			27.5 (14)		
	>95th	20.8 (11)	0.8	0.3–1.8	24.5 (13)	1.4	0.6–3.4	26.2 (16)	0.9	0.4–2.2
dlPCB	≤95th	22.9 (11)			19.2 (10)			33.3 (18)		
	>95th	23.8 (15)	1.1	0.4–2.6	23.7 (14)	1.3	0.5–3.3	20.7 (12)	0.5	0.2–1.2
Cortisol ↓	≤95th	7.1 (2)			0.0 (0)			2.1 (1)		
LPCB	>95th	3.6 (3)	0.5	0.1–3.1	2.6 (2)	- ^1^	-	1.6 (1)	0.8	0.1–12.2
	≤95th	5.2 (3)			0.0 (0)			3.9 (2)		
HPCB	>95th	3.8 (2)	0.7	0.1–4.5	3.8 (2)	- ^1^	-	0.0 (0)	- ^1^	-
	≤95th	6.3 (3)			0.0 (0)			3.7 (2)		
dlPCB	>95th	3.2 (2)	0.5	0.1–3.1	3.4 (2)	- ^1^	-	0.0 (0)	- ^1^	-
	≤95th	7.1 (2)			0.0 (0)			2.1 (1)		

*Notes:* PCB = polychlorinated biphenyls; LPCB = lower-chlorinated PCB; HPCB = higher-chlorinated PCB; dlPCB = dioxin-like PCB; DHEAS = dehydroepiandrosterone sulfat; t1–t3 = sampling time point 1–3; PC = percentile; *n* = frequency; OR = odds ratio; CI = confidence interval; ↑ = above reference range; ↓ = below reference range; abnormal = above or below reference range. Significant ORs are in bold. ^1^ No odds ratio can be calculated, because there are no cases in one cell.

**Table 5 ijerph-17-04708-t005:** Comparison of mean stress hormone concentrations between normal and higher-burdened participants (*N* = 112).

	t1						t2						t3					
	**PCB Body Burden ^1^**						**PCB Body Burden ^1^**						**PCB Body Burden ^1^**					
	**≤95th**	**>95th**	**F**	**df**	***p***	**η²**	**≤95th**	**>95th**	**F**	**df**	***p***	**η²**	**≤95th**	**>95th**	**F**	**df**	***p***	**η²**
	**M (S.E.)**	**M (S.E.)**					**M(SD)**	**M(SD)**					**M (SD)**	**M (SD)**				
LPCBs																		
DHEAS	7.0 (0.7)	8.1 (0.4)	2.2	1	0.07	0.02	**6.3 (0.6)**	**7.7 (0.4)**	**3.5**	**1**	**0.03**	**0.03**	**6.0 (0.5)**	**7.4 (0.4)**	**4.3**	**1**	**0.02**	**0.04**
Cortisol	411.4 (31.4)	416.1 (18.1)	0.2	1	0.45	0.00	447.6 (23.6)	441.0 (15.4)	0.1	1	0.41	0.00	427.0 (20.5)	430.1 (17.4)	0.0	1	0.46	0.00
HPCBs																		
DHEAS	7.6 (0.5)	8.1 (0.5)	0.5	1	0.25	0.01	6.9 (0.5)	7.6 (0.5)	1.3	1	0.13	0.01	6.4 (0.5)	7.1 (0.4)	1.1	1	0.15	0.01
Cortisol	416.7 (22.0)	413.0 (22.9)	0.1	1	0.46	0.00	443.5 (18.0)	442.4 (18.0)	0.0	1	0.49	0.00	416.4 (19.8)	439.2 (18.1)	0.7	1	0.21	0.01
dlPCBs																		
DHEAS	7.4 (0.5)	8.2 (0.5)	1.6	1	0.11	0.02	**6.6 (0.5)**	**7.8 (0.5)**	**3.3**	**1**	**0.04**	**0.03**	6.4 (0.5)	7.2 (0.4)	1.5	1	0.12	0.01
Cortisol	416.1 (24.0)	414.0 (21.3)	0.0	1	0.48	0.00	442.3 (19.0)	443.6 (18.1)	0.0	1	0.48	0.00	427.2 (19.3)	430.3 (18.8)	0.0	1	0.46	0.00

Notes: Controlled for age, albumin and smoking; t1–t3 = sampling time point 1–3; PCB = polychlorinated biphenyls, LPCB = lower chlorinated PCBs, HPCB = higher chlorinated PCBs, dlPCB = dioxin-like PCBs; DHEAS = dehydroepiandrosterone sulfate; M = mean, SD = standard deviation; F = F-value; df = degrees of freedom, *p* = *p*-value (significance); η² = ETA² (effect size). ^1^ ng/g lipid. Significant results are in bold.

**Table 6 ijerph-17-04708-t006:** Cross-sectional associations and associations over time between PCBs and DHEAS and cortisol from multiple linear regressions analyses (*N* = 112).

	t1				t2				t3			
	ß	T	*p*	R²	ß	T	*p*	R²	ß	T	*p*	R²
DHEAS												
LPCB_t1	0.08	0.86	0.20	0.32	−0.17	−0.68	0.25	0.35	0.06	0.22	0.42	0.22
LPCB_t2					**0.17**	**2.00**	**0.03**	**0.35**	0.27	0.76	0.23	0.20
LPCB_t3									0.10	1.06	0.15	0.22
HPCB_t1	0.05	00.54	0.30	0.31	−0.34	−0.78	0.22	0.34	0.34	0.82	0.21	0.22
HPCB_t2					0.10	1.21	0.12	0.33	0.16	0.36	0.36	0.20
HPCB_t3									0.07	0.71	0.24	0.22
dlPCB_t1	0.06	0.74	0.23	0.31	−0.42	−0.92	0.18	0.34	0.27	0.51	0.31	0.22
dlPCB_t2					0.13	1.55	0.07	0.34	0.09	0.15	0.44	0.20
dlPCB_t3									0.09	0.99	0.16	0.22
OHPCB_t1	0.06	0.71	0.24	0.29	0.20	0.90	0.17	0.31	0.03	0.11	0.46	0.18
OHPCB_t2					0.09	1.05	0.15	0.31	−0.03	−0.12	0.46	0.17
OHPCB_t3									0.07	0.72	0.24	0.20
Cortisol												
LPCB_t1	−0.01	−0.09	0.47	0.02	−0.51	−1.63	0.06	0.06	−0.05	−0.17	0.44	0.00
LPCB_t2					0.07	0.70	0.25	0.05	0.33	0.85	0.20	0.04
LPCB_t3									0.02	0.24	0.42	0.07
HPCB_t1	−0.11	-1.06	0.15	0.03	−0.79	−1.52	0.07	0.06	−0.25	−0.53	0.30	0.01
HPCB_t2					0.02	0.23	0.41	0.04	−0.08	−0.17	0.44	0.04
HPCB_t3									0.02	0.16	0.44	0.07
dlPCB_t1	−0.08	−0.82	0.21	0.03	−0.45	−0.83	0.21	0.04	−0.28	−0.49	0.32	0.00
dlPCB_t2					0.01	0.13	0.45	0.04	−0.25	−0.38	0.36	0.04
dlPCB_t3									0.03	0.33	0.38	0.07
OHPCB_t1	−0.10	−0.95	0.17	0.04	−0.24	−0.90	0.19	0.04	0.26	0.92	0.18	0.01
OHPCB_t2					0.03	0.33	0.37	0.05	−0.29	−1.06	0.15	0.05
OHPCB_t3									−0.01	−0.09	0.47	0.06

Notes: Controlled for age, albumin and smoking; ß = standardized regression coefficient, T = T value of regression coefficients, R² = explained variance, t1–t3 = sampling time point 1–3; PCB = polychlorinated biphenyls, NPCB = lower chlorinated PCBs, HPCB = higher chlorinated PCBs, dlPCB = dioxin-like PCBs, OHPCB = hydroxylated PCBs. Cross-sectional associations are in the grey field. Significant results are in bold.

**Table 7 ijerph-17-04708-t007:** Fixed and random effects of the association between PCBs and the stress hormone concentrations of DHEAS and cortisol (*N* = 112).

		LPCBs			HPCBs			dlPCBs			OH-PCBs		
		β	t	*p*	β	t	*p*	β	t	*p*	β	t	*p*
**DV: DHEAS**													
**Fixed effect**	PCBs	**0.09**	**1.83**	**0.04**	**0.08**	**1.67**	**0.05**	0.08	1.59	0.06	0.08	1.50	0.07
	Age	−0.50	−8.51	0.00	−0.52	−9.48	0.00	−0.51	1.59	0.00	−0.50	−8.47	0.00
	Albumin	0.13	2.59	0.01	0.13	2.65	0.01	0.12	−8.98	0.01	0.13	2.53	0.01
	Smoking	−0.02	−0.43	0.34	−0.03	−0.51	0.61	−0.02	2.59	0.66	−0.02	−0.032	0.38
	R²	0.29			0.29			0.29			0.27		
**Random effect ^1^**	Variance	0.00			0.00			0.00			0.00		
	Standard deviation	0.00			0.00			0.00			0.00		
	R²	0.00			0.00			0.00			0.00		
**DV: Cortisol**													
**Fixed effect**	PCBs	−0.00	−0.01	0.45	−0.02	−0.30	0.39	−0.03	−0.45	0.33	−0.02	−0.30	0.38
	Age	0.03	0.65	0.26	0.04	0.66	0.25	0.04	0.56	0.29	0.04	0.58	0.28
	Albumin	−0.01	4.15	0.00	0.22	4.10	0.00	0.23	4.11	0.00	0.24	4.21	0.00
	Smoking	−0.01	−1.03	0.15	−0.06	−0.95	0.17	−0.06	−0.94	0.35	−0.04	−0.70	0.25
	R²	0.05			0.05			0.05			0.06		
**Random effect ^1^**	Variance	0.003			0.003			0.003			0.002		
	Standard deviation	0.06			0.06			0.06			0.04		
	R²	0.003			0.003			0.004			0.002		

Notes: ^1^ random effect for sampling time point; PCBs = polychlorinated biphenyls, LPCBs = lower-chlorinated Biphenyls, HPCBs = higher-chlorinated PCBs; dlPCBs = dioxin-like PCBs, OH-PCBs = hydroxylated PCBs; β = standardized regression coefficient, t = t-value, *p* = *p*-value (significance), R² = R squared (explained variance). Significant results are in bold.

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
