# Peer review of "Cross-Sectional and Longitudinal Effects of PCB Exposure on Human Stress Hormones in the German HELPcB Surveillance Program"

_ijerph, 2020, doi:10.3390/ijerph17134708_

Round 1
Reviewer 1 Report
This manuscript describes a longitudinal study on the associations between plasma concentrations of PCBs and stress hormones DHEAS and cortisol in a cohort of 131 former e-waste recycling workers.
General comments
There are inconsistencies in the reference numbering between the text and the reference list. UNEP has no number in the reference list resulting in erroneous numbering of the following references. In the Introduction, the general endocrine disrupting properties of PCBs could be mentioned. See, The State of the Science of Endocrine Disrutping Chemicals, UNEP/WHO 2012.
Specific comments
- 38: Avoid “estrogenic” PCBs as it is not defined here.
- 41: It would be more relevant to state that the ban on use of PCBs came into force in Germany in 1989.
- 47: PCBs with the same number of Cl-substituents are structural isomers, not stereoisomers and the sentence after i.e. is wrong. The authors are confusing this with chiral PCBs that have two enantiomeric forms.
- 49: The authors should already here define what they mean by low and high chlorinated PCBs. This is not a standard definition.
- 56: Why are the dl-PCBs mentioned here for inhalation and dermal uptake in work places? They comprise biphenyls with 4-7 Cl, i.e., both “low” and “high” chlorinated PCBs.
- 75 f.: between PCB exposure (or body burden) and the stress hormone…
- 82: Concentrations of PCBs in plasma/serum reflects both the historic exposure and the present body burden. I recommend using the term background exposure. Further, the term “normal burdened” is very unusual and should be replaced by background exposed. Corresponding changes should be made throughout the manuscript.
- 93: I recommend to substitute “measurement occasions” with “sampling time points”.
- 94: Rephrase sentence in particular “stable over time”.
- 104: The authors should specify the type of recycling plant and work done by the participants. Further, information on the time lag since the work exposure should be given.
- 115: Plasma is first deproteinized and then extracted.
- 116: I guess there is a clean-up step prior to GC-MS.
- 117: …was analysed by using online solid phase extraction coupled to…
- 127: …, but only the eight mono-ortho dl-PCBs could be detected in more…
The four non-ortho dl-PCBs are usually determined by a Dioxin method.
- 137: The substitution with a hydroxyl group does not significantly alter the lipophilicity, particularly in the high chlorinated PCBs (Dakai et al. Environ Sci Pollut Res Int. 2018;25:16277–16290.
- 152: “influencing factors” Do the authors mean confounding factors?
Table 2: I did not find a definition of t1, t2 and t3.
- 219: The PCB levels for the highly exposed and background exposed groups should be defined. Further, how many participants are in each group?
- 215: This paragraph is a repetition of the last part of the Introduction and should be omitted.
- 294: …which is the reason..
- 304: This sentence is unclear. Only few PCB congeners were determined and structural isomers with different substitution pattern are not compared.
- 306: This sentence is unclear. Rephrase.
- 311: The sentence is not quite clear. Rephrase.
- 345: …the mechanism behind the positive association between PCB exposure and DHEAS increase in plasma.
Author Response
Dear Reviewer 1,
we thank you very much for your helpful comments regarding our submission. We have revised the manuscript based on the valuable suggestions and advice you made. An item-by-item response to your comments is enclosed. Please note, all page and line numbers refer to the document with the markups, so you can track the changes. We hope that these revisions successfully address your concerns and requirements.
We thank you for your helpful suggestions and your time.
Sincerely,
Dr. Petra Gaum
Comments of reviewer 1 and our response (in italic):
This manuscript describes a longitudinal study on the associations between plasma concentrations of PCBs and stress hormones DHEAS and cortisol in a cohort of 131 former e-waste recycling workers.
General comments
There are inconsistencies in the reference numbering between the text and the reference list. UNEP has no number in the reference list resulting in erroneous numbering of the following references. In the Introduction, the general endocrine disrupting properties of PCBs could be mentioned. See, The State of the Science of Endocrine Disrutping Chemicals, UNEP/WHO 2012.
Thank you for this hint. We checked all references and have seen that no number was given to one reference. We corrected this and adopted the reference list to the new order of the revised manuscript.
Yes, you are right. It would be possible to describe PCBs in general as endocine disruptors, but reviewer 2 noted that more attention should be paid to health effects of PCB exposure during the whole manuscript. Therefore, we decided not to condense the described health effects in the introduction.
Specific comments
- 38: Avoid “estrogenic” PCBs as it is not defined here.
Thank you. That's a good hint. We deleted the term "estrogenic." See P. 2 L. 39.
- 41: It would be more relevant to state that the ban on use of PCBs came into force in Germany in 1989.
We have added the information in the introduction. See P. 2 L. 44.
- 47: PCBs with the same number of Cl-substituents are structural isomers, not stereoisomers and the sentence after i.e. is wrong. The authors are confusing this with chiral PCBs that have two enantiomeric forms.
Thank you very much for this hint. We reworded the sentence and hope that it contains the correct information now. See P. 2 L. 47-54.
- 49: The authors should already here define what they mean by low and high chlorinated PCBs. This is not a standard definition.
Now, we define the created groups in more detail and also deal more explicitly with the fact that dlPCBs can be both; lower- and higher-chlorinated (also in relation to your next comment). See P. 2 L.52-54.
- 56: Why are the dl-PCBs mentioned here for inhalation and dermal uptake in work places? They comprise biphenyls with 4-7 Cl, i.e., both “low” and “high” chlorinated PCBs.
Yes, you are right. Only lower-chlorinated dl-PCBs are typical for inhalation and dermal uptake. Both were mentioned because of work-related exposure. Now, we describe different paths of exposure and specified the information in the brackets. See P.2 L. 57-63.
- 75 f.: between PCB exposure (or body burden) and the stress hormone…
Now, we used the term PCB "body burden", which best describes our variable. See P. 3 L. 84.
- 82: Concentrations of PCBs in plasma/serum reflects both the historic exposure and the present body burden. I recommend using the term background exposure. Further, the term “normal burdened” is very unusual and should be replaced by background exposed. Corresponding changes should be made throughout the manuscript.
To our knowledge background exposure refers to an environmental exposure that affects everybody in an environmental area, but we have an occupational exposure. Nevertheless, we use the term „background exposed”, because it’s a common term and we define what we mean by background burden in the Method section under statistical analyses. See P. 6 L. 181-185.
- 93: I recommend to substitute “measurement occasions” with “sampling time points”.
Now we use the term “sampling time points” instead of measurement occasion during the whole manuscript.
- 94: Rephrase sentence in particular “stable over time”.
We rephrased the mentioned part of the sentence into „…continuously over all three sampling time points”. See P. 3 L. 102f.
- 104: The authors should specify the type of recycling plant and work done by the participants. Further, information on the time lag since the work exposure should be given.
We specified the type of work and the recycling plant our participant worked until 2010. See P. 3 L. 113-115.
- 115: Plasma is first deproteinized and then extracted.
We included a detailed description of the PCB and OH-PCB analyses in the appendix of the manuscript (see Appendix A). We included the information in the appendix to ensure the flow of reading.
- 116: I guess there is a clean-up step prior to GC-MS.
Please see our answer for the comment before.
- 117: …was analysed by using online solid phase extraction coupled to…
Please see our answer for the comment before.
- 127: …, but only the eight mono-ortho dl-PCBs could be detected in more…
We rephrased the sentence according to your suggestion. See P.2 L. 141-143.
The four non-ortho dl-PCBs are usually determined by a Dioxin method.
Yes, you are right. This is the reason why we could only detect the eight mono-ortho-dl-PCBs.
137: The substitution with a hydroxyl group does not significantly alter the lipophilicity, particularly in the high chlorinated PCBs (Dakai et al. Environ Sci Pollut Res Int. 2018;25:16277–16290.
Yes this is correct, but in the plasma OH-PCBs conjugate with glucoronide or sulfate in the liver and thus are not as lipophilic as the parent congeners. In included the following information in the method section; see P 6 L152-154.
152: “influencing factors” Do the authors mean confounding factors?
Yes, we meant confounding factors and adapted the term. See P. 6 L 169.
Table 2: I did not find a definition of t1, t2 and t3.
We have seen that the explanation of “t” was missing in Table 1. We included the definition there and specified the definition of t1-t3 in all table notes.
- 219: The PCB levels for the highly exposed and background exposed groups should be defined. Further, how many participants are in each group?
We included further information about the created groups. Specifically, the number of participants in each group and their PCB body burden. Please see Table 1b (P. 5).
- 215: This paragraph is a repetition of the last part of the Introduction and should be omitted.
Do you mean line 255ff? Yes, the part from line 255ff is nearly the same as at the End of the introduction. We omitted it. Please see P. 9. L. 275-280.
- 294: …which is the reason..
Thank you for this hint. We changed the sentence as suggested. See P. 13. L. 314.
- 304: This sentence is unclear. Only few PCB congeners were determined and structural isomers with different substitution pattern are not compared.
Thank you very much for this hint. There was a part missing in this sentence. We added the missing part and hope that the sentence is clearer now. See P. 13 L. 323-326.
- 306: This sentence is unclear. Rephrase.
We have rewritten the sentence and hope that it is easier to understand now; also with regard to the correction of the previous sentence (see our answer to your comment before). See P. 13 L. 326-331.
- 311: The sentence is not quite clear. Rephrase.
We restructured the paragraph and extended it. Hopefully the whole paragraph including the mentioned sentence is clearer now. See P. 13 L. 332-340.
- 345: …the mechanism behind the positive association between PCB exposure and DHEAS increase in plasma.
We rephrased the mentioned sentence according to your suggestion. See P. 14. L. 381f.
Reviewer 2 Report
Review of manuscript: Cross-sectional and longitudinal effects of PCB exposure on human stress hormones in the German HELPcB surveillance program.
The manuscript describes cross-sectional and longitudinal effects of PCB exposure on the stress hormones DHEAS and cortisol.
Although the topic of the paper appears to be interesting, the authors should focus on the improvement of the article. Here is my opinion:
- Is this correct: Eight further participants (0,03%) – if 8 individuals is 0,03%, how many participants should you have then?
- Do the authors think that the geographic place of living of peopel could affect the results? If yes, how? Please add a few sentences in introduction.
- Please, check the manuscript for typos.
- The authors should also provide on the validation of the method (e.g. linearity, accuracy, precision etc.). Please add those data.
- Can PCB exposure quantified somehow?
- Health consequences of prolonged PCB exposure should be discussed more thoroughly to improve the importance of the paper.
- The authors should provide a paragraph on the possible application of their studies. How these results can affect the environment or science?
- Check please the References: However, Uh et al. also found a non-linear correlation between the DHEAS concentration and the occurrence of depression of varying severity [42]. But 42 is Lee et al. The same with the next position. You have 47 ref in the text, but 46 in the reference list.
Author Response
Dear Reviewer 2,
we thank you very much for your helpful comments regarding our submission. We have revised the manuscript based on the valuable suggestions and advice you made. An item-by-item response to your comments is enclosed. Please note, all page and line numbers refer to the document with the markups, so you can track the changes. We hope that these revisions successfully address your concerns and requirements.
We thank you for your helpful suggestions and your time.
Sincerely,
Dr. Petra Gaum
Comments of reviewer 1 and our response (in italic):
The manuscript describes cross-sectional and longitudinal effects of PCB exposure on the stress hormones DHEAS and cortisol.
Although the topic of the paper appears to be interesting, the authors should focus on the improvement of the article. Here is my opinion:
- Is this correct: Eight further participants (0,03%) – if 8 individuals is 0,03%, how many participants should you have then?
Thank you very much for your hint. In fact, it was forgotten to multiply by 100 here. The correct number is 2.67%. We corrected the number; see P. 3 L. 120.
- Do the authors think that the geographic place of living of peopel could affect the results? If yes, how? Please add a few sentences in introduction.
Yes, the place of living affect the PCB exposure. We included this information in the introduction. See P. 2 L. 57-59.
All participants of our study cohort came from the same area in Germany. The probability of differences in PCB-Exposure and a bias of the results due to geographical differences is very low.
- Please, check the manuscript for typos.
Thank you very much for this hint. We carefully read the entire manuscript again and hope that all typing errors have been corrected now. Among other things, missing commas have been replaced (e.g. P. 3 L. 98).
- The authors should also provide on the validation of the method (e.g. linearity, accuracy, precision etc.). Please add those data.
We included a detailed description of the PCB and OH-PCB analyses in the appendix of the manuscript including the validation of the method (see Appendix A). We included the information in the appendix to ensure the flow of reading.
- Can PCB exposure quantified somehow?
Table 1a shows the descriptive values for PCB exposure. We also added another table that shows the descriptive parameters of each congener (see Appendix B).
- Health consequences of prolonged PCB exposure should be discussed more thoroughly to improve the importance of the paper.
Now, we focused on the information about illnesses following PCB exposure throughout the document. In the introduction and discussion section, we pay attention to the carcinogenicity of PCBs and on a possible link to increased DHEAS concentrations. We hope that this enhance perception of the relevance of this manuscript and its findings.
Please see P. 2 L. 64-76 in the Introduction and P. 15 L.340-352 in the discussion section.
- The authors should provide a paragraph on the possible application of their studies. How these results can affect the environment or science?
Now, we discuss more about the benefits of the study results and recommend further research on this area in the discussion section. See P. 13 L. 341-364.
- Check please the References: However, Uh et al. also found a non-linear correlation between the DHEAS concentration and the occurrence of depression of varying severity [42]. But 42 is Lee et al. The same with the next position. You have 47 ref in the text, but 46 in the reference list.
Thank you for this hint. We checked all references and have seen that no number was given to one reference. We corrected this and adopted the reference list to the new order of the revised manuscript.
Round 2
Reviewer 2 Report
The authors addressed all my suggestions.